# The $\omega^3$ scaling of the vibrational density of states in quasi-2D nanoconfined solids

Yuanxi Yu [1,2,15], Chenxing Yang [1,2,15], Matteo Baggioli [3,4,16 ✉], Anthony E. Phillips [5], Alessio Zaccone [6,7], Lei Zhang [2,8], Ryoichi Kajimoto [9], Mitsutaka Nakamura [9], Dehong Yu [10] & Liang Hong [1,2,11,12,13,14,16 ✉]

The vibrational properties of crystalline bulk materials are well described by Debye theory, which successfully predicts the quadratic $\omega^2$ low-frequency scaling of the vibrational density of states. However, the analogous framework for nanoconfined materials with fewer degrees of freedom has been far less well explored. Using inelastic neutron scattering, we characterize the vibrational density of states of amorphous ice confined inside graphene oxide membranes and we observe a crossover from the Debye $\omega^2$ scaling to an anomalous $\omega^3$ behaviour upon reducing the confinement size $L$. Additionally, using molecular dynamics simulations, we confirm the experimental findings and prove that such a scaling appears in both crystalline and amorphous solids under slab-confinement. We theoretically demonstrate that this low-frequency $\omega^3$ law results from the geometric constraints on the momentum phase space induced by confinement along one spatial direction. Finally, we predict that the Debye scaling reappears at a characteristic frequency $\omega_\times = vL/2\pi$, with $v$ the speed of sound of the material, and we confirm this quantitative estimate with simulations.

[1] School of Physics and Astronomy, Shanghai Jiao Tong University, 200240 Shanghai, China. [2] Institute of Natural Sciences, Shanghai Jiao Tong University, 200240 Shanghai, China. [3] Wilczek Quantum Center, School of Physics and Astronomy, Shanghai Jiao Tong University, 200240 Shanghai, China. [4] Shanghai Research Center for Quantum Sciences, 201315 Shanghai, China. [5] School of Physics and Astronomy, Queen Mary University of London, London, UK. [6] Department of Physics "A. Pontremoli", University of Milan, via Celoria 16, 20133 Milan, Italy. [7] Cavendish Laboratory, University of Cambridge, CB3 0HE Cambridge, UK. [8] School of Materials Science and Engineering, Shanghai Jiao Tong University, 200240 Shanghai, China. [9] J-PARC Center, Japan Atomic Energy Agency (JAEA), Tokai, Ibaraki 319-1195, Japan. [10] Australian Nuclear Science and Technology Organisation, Lucas Heights, NSW 2234, Australia. [11] Shanghai National Center for Applied Mathematics (SJTU Center), Shanghai Jiao Tong University, 200240 Shanghai, China. [12] Shanghai Artificial Intelligence Laboratory, 200232 Shanghai, China. [13] School of Medicine, Shanghai Jiao Tong University, 200240 Shanghai, China. [14] Zhangjiang Institute for Advanced Study, Shanghai Jiao Tong University, 200240 Shanghai, China. [15] These authors contributed equally: Yuanxi Yu, Chenxing Yang. [16] These authors jointly supervised this work: Matteo Baggioli, Liang Hong. ✉email: b.matteo@sjtu.edu.cn; hongl3liang@sjtu.edu.cn

Describing the vibrational and thermodynamic properties of matter is one of the long-standing tasks of solid-state physics[1]. In the case of crystalline bulk solids, the problem has been solved a long-time ago, in 1912, by Peter Debye with his celebrated model[2]. Debye's theory correctly predicts the quadratic $\sim \omega^2$ low-frequency scaling of the vibrational density of states (VDOS) of bulk solids and the corresponding $\sim T^3$ low-temperature scaling of their specific heat, in perfect agreement with many experimental observations[1,3].

The Debye model relies only on two fundamental assumptions: (I) the low-energy vibrational dynamics is governed by a set of propagating Goldstone modes[4], i.e. acoustic phonons, with linear dispersion relation $\omega = v k$ and (II) the phase space for the allowed wavevectors is given by a perfect spherical manifold of radius $k_D$, the Debye wavevector. The first assumption is notably violated in liquids in which the late-time (low energy) dynamics is dominated by diffusion[5], which induces a constant contribution to the density of states observed both in molecular dynamics simulations and in experiments. Additionally, in liquids, normal modes coexist at low energy with a large number of unstable instantaneous normal modes (INMs) which appear as negative eigenvalues of the Hessian or dynamical matrix and they reflect the presence of numerous saddle points in the highly anharmonic potential landscape[6–8]. Consequently, even when the diffusive contribution is neglected, the density of states in liquids grows linearly with the frequency instead of quadratically as predicted by Debye theory[9]. At the same time, the specific heat does not grow with temperature but rather decreases[10]. Both these peculiar behaviours, which remarkably deviate from Debye's theory, can be explained in terms of additional low energy degrees of freedom, the unstable INMs[11,12]. In a nutshell, paraphrasing Stratt[7], the deviations from Debye's theory in liquids are simply due to the fact that "liquids are not held together by springs," and strong anharmonicities play a dominant role. The first assumption of Debye's theory is also notably violated in glasses, which present additional quasi-localized modes and exhibit well-known anomalies in the vibrational and thermodynamic properties with respect to their ordered counterparts[13].

Despite the known examples of amorphous systems (liquids and glasses), whether the Debye model, and indeed even the continuum theory of elasticity, work in ordered solids under strong spatial confinement is largely unknown. As we will show, geometric confinement could indeed lead to a violation of the assumption (II) presented above and a deviation from the Debye scaling even in ordered solids, without the need of having additional low energy modes as in liquids or glasses.

Nanometer confinement is ubiquitous in the frontiers of biotechnology, electronic engineering, and material sciences to achieve unprecedented advantage, e.g. applications of graphene or graphene-based materials for electron conduction[14–16], seawater desalination[17,18] and biosensing[19,20]. Atomic vibrations in confined environments play a crucial role in a plethora of phenomena such as facilitating electron conduction in graphene-based electronic devices[21,22], enhancing proton delivery through the ion channel across the cell membrane[23,24] and conducting energy via the power enzymatic motion of protein molecules[25,26]. The effects of strong confinement have been investigated in nanoconfined liquids[27–31] where liquid-to-solid transitions have been found[32–34]. More importantly, the role of confinement (specially in nanopores geometries with confinement along two spatial directions)in the VDOS of glasses have been discussed in several works with particular attention to the fate of the boson peak anomaly and to the effects on the glassy relaxational dynamics. In the case of hard confinement, when the dynamics of the confining matrix is slower than that of the confined material, an ubiquitous reduction of the low-frequency part of the VDOS

below the boson peak frequency has been observed experimentally in polymers[35], metallic glasses[36], glass-forming liquid salol[37], molecular glass former dibutyl phthalate/ferrocene[38] and liquid crystals E7[39]. In the alternative scenario of soft confinement, the behavior is opposite and the weight is shifted towards lower frequencies upon confinement, as shown experimentally in propylene glycol[40], discotic liquid crystals[41]. In the context of amorphous solids, the difference between soft and hard confinement and the importance of the boundary conditions have been established in[42]. As we will see, our findings are in qualitative agreement with those of ref. [42]. We refer the reader to ref. [43] for a comprehensive review of the influence of spatial confinement on the dynamics of glass-forming systems. Finally, a crossover between 2D and 3D Debye law has been observed experimentally in gold nanostructures[44] and more recently in MD simulations data[45]. Although the importance of reduced dimensionality to nanoscience has long been appreciated, most of the previous vibrational studies have focused on 0D or 1D confinement in nanopores and a qualitative deviation from Debye law at low frequency has never been observed. To the best of our knowledge, the effects of nano-confinement on the VDOS of solids confined in slab geometries have not been considered so far and they are indeed the topic of this manuscript.

In this work, we performed both inelastic neutron scattering and molecular dynamics simulations on quasi-2D nanoconfined crystalline and amorphous ice where the length scale of confinement can be well controlled between 7 Å and 20 Å. We found that the low-energy vibrational density of states of the confined solids exhibits a robust $\sim \omega^3$ power law, faster than the expected $\sim \omega^2$ Debye's law (see Fig. 1 for a schematic comparison between the Debye's result, the linear behavior in liquids and the case of confined solids considered in this work). We further show with a simple analytical model that the deviation from Debye's law arises from the geometrical constraints on the available wavevector phase space, without any changes in the nature of the low-energy excitations, which are still well-described by propagating plane waves $\omega = vk$ as demonstrated by auxiliary molecular dynamics simulations. Finally, combining molecular dynamic simulations and theoretical arguments, we have been able to show that the crossover between the low-frequency $\omega^3$ scaling and the standard Debye law appears at a characteristic wavevector $k_\times = 2\pi/L$, with $L$ the size of the confined direction. Despite similar studies[35–39,41–43,46], claiming a low-energy suppression of the VDOS due to confinement, have been performed in nano-confined amorphous systems, this scaling and its theoretical foundations have been never mentioned nor discussed in the past.

## Results

**Experimental evidence.** Using inelastic neutron scattering, we have measured the VDOS of water at 120 K (solid state) softly confined[47] between graphene oxide membranes (GOM) with different levels of hydration. The detailed information on the sample preparation, the experimental methods and the data analysis are provided in the Methods.

As a reference, in Fig. 2a, we present the inelastic neutron scattering data of the bulk crystalline ice measured in this work and those for bulk high-density amorphous ice (HDA) and bulk low-density amorphous ice (LDA) ice taken from ref. [48]. At low frequency, approximately below 4 meV, we observe a leveling-off of the reduced density of states, $g(\omega)/\omega^2$, and a well-defined Debye level, in agreement with the standard Debye model expectation, $g(\omega) \sim \omega^2$. This is further supported by the VDOS of bulk crystalline and disordered ice samples derived from MD simulations (see next section). Hence, the Debye model is a valid description of the low-energy vibrational dynamics in bulk solids, as expected. In sharp contrast, in the sample with lowest

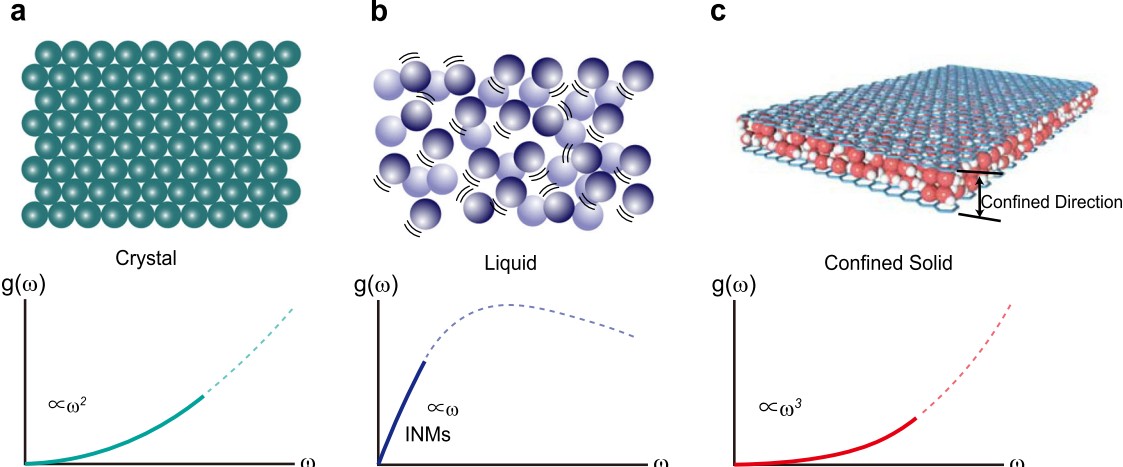

**Fig. 1 The low-frequency vibrational density of states(VDOS) for three different kinds of systems.** The (low frequency) VDOS $g(\omega)$ for ordered crystalline solids with long-range order (**a**), liquids (**b**) and slab-confined solids (**c**). Ordered solids, at low frequency, display a characteristic $\omega^2$ scaling as predicted by Debye theory. Liquids, once the diffusive contribution is removed, shows a peculiar linear scaling due to the presence of unstable instantaneous normal modes. As demonstrated in this work, slab-confined solids, whether amorphous or ordered, exhibits a $\omega^3$ scaling which is visible both in experiments and simulations and it can be explained analytically by considering simple geometric constraints on the wavevector phase space of low-energy phonon modes.

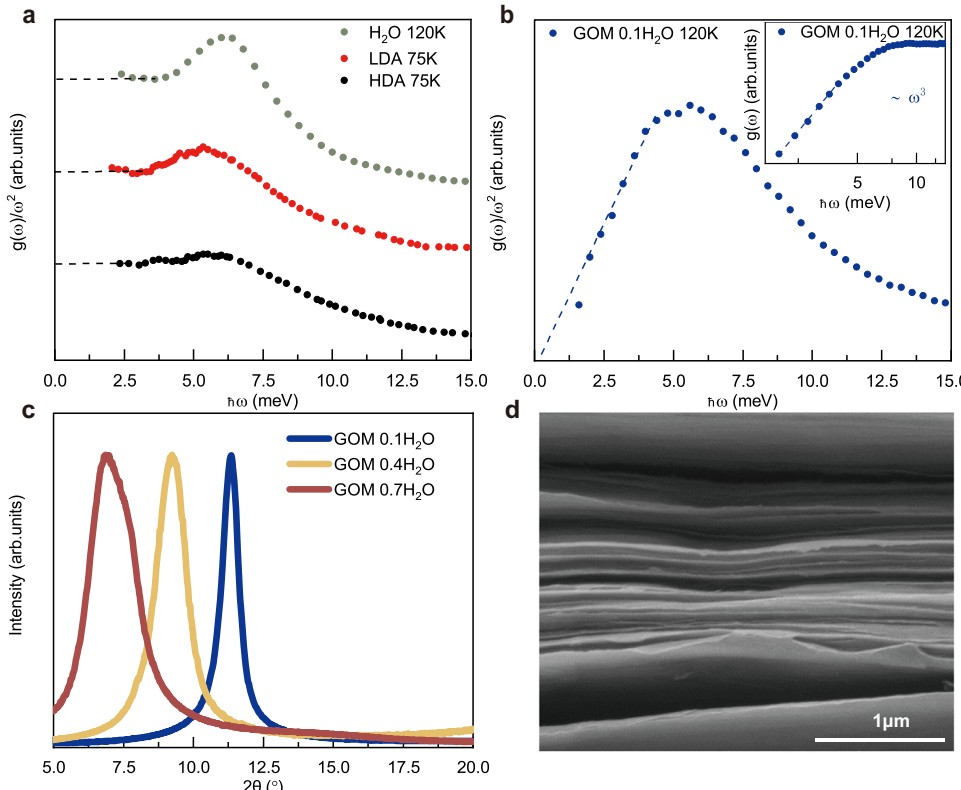

**Fig. 2 The $\omega^3$ scaling induced by the confinement of graphene oxide membrane(GOM). a** The Debye normalized VDOS $g(\omega)/\omega^2$ for high density amorphous (HDA) ice, low density amorphous ice (LDA) obtained from ref. [48] and bulk crystalline ice measured in this work at 120K. The horizontal dashed lines indicate the Debye level for each sample. The curves have been manually shifted vertically for better presentation and the vertical axes has arbitrary units. **b** The experimental normalized VDOS $g(\omega)/\omega^2$ for the hydrated GOM sample at $h = 0.1$ g water/gram GOM. The dashed line indicates the low frequency $\sim\omega^3$ scaling due to confinement. The inset shows the original VDOS data in log–log scale. The $y$-axes are presented in arbitrary units. **c** The XRD data of GOM sample with different hydration levels. **d** The scanning electron microscope (SEM) image of the GOM sample which highlights the fuzzy boundaries of the graphene oxide membranes.

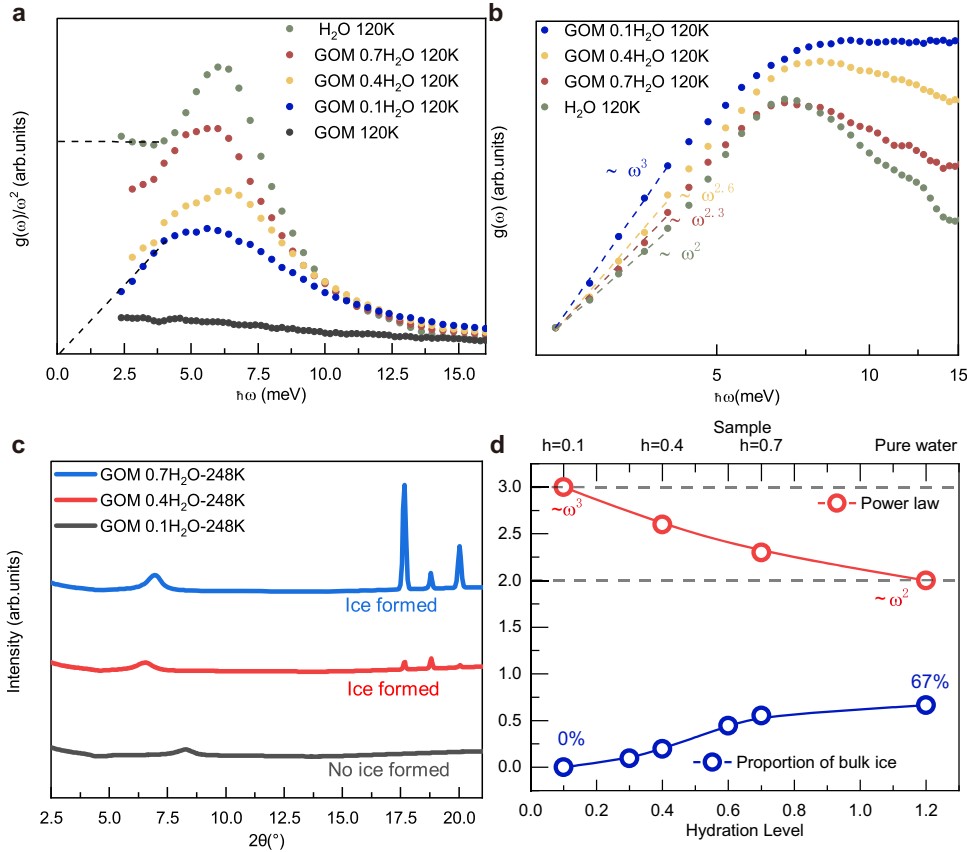

**Fig. 3 The evolution of VDOS for confined ice upon changing the confiment size. a** The experimental VDOS normalized by $\omega^2$ vs. frequency for bulk ice (green), GOM sample at $h = 0.7$ (red), 0.4 (yellow), 0.1 (blue) and the dry GOM (gray), respectively. The horizontal dashed line guides the eyes towards the Debye level for the bulk sample. The diagonal dashed line guides the eye towards the $g(\omega) \sim \omega^3$ low frequency scaling. The blue data correspond to those reported in the panel (b) of Fig. 2. **b** The log-log scale plot of the original non-normalized VDOS corresponds to the data in panel (**a**). All datasets are fitted in the same energy interval (from 2.4 to 4 meV) and rescaled such that the first data points overlap for better comparison. **c** The X-ray diffraction data for samples at different hydration levels. **d** The fitting power of the low frequency VDOS data for different hydration levels (red). The dashed lines indicate the $\omega^3$ scaling and the standard quadratic Debye's law. The DSC data showing the mass proportion of bulk ice for the corresponding hydration levels (blue).

hydration level (0.1 g water per gram GOM), where the distance between the neighboring GOM layers is only 7Å (Fig. 2c), the reduced density of states displays a neat low-frequency linear scaling, thus the original VDOS scales as $g(\omega) \sim \omega^3$ (Fig. 2b). The same feature is shown in logarithmic scale in the inset of Fig. 2b. This cubic regime appears robust in a large range of frequencies from ≈1.6 meV to ≈5 meV. Below 1.6 meV, the measurements are affected by the resolution function of the instrument (0.35 meV upwards to 1 meV or so ref. [49]) and thus the corresponding data are not presented. In order to make sure that this anomalous scaling does not stem from the GOM structure itself, we measured the VDOS of the dry GOM without any water in between (shown in Fig. 3a). The resulting reduced VDOS, presented in Fig. 3a, slowly decreases with frequency and does not present any relevant features. Based on the dependence of the reduced VDOS on the frequency $\omega$ and the absolute amplitude of the dry sample, one can deduce that the observed anomalous $\omega^3$ scaling in hydrated GOM must result from the nano-confined water. Further X-ray diffraction measurements reveal that the water confined in GOM at this hydration level is amorphously packed as no characteristic Bragg peak of the crystalline ice is present (see Fig. 3c). At such low temperature, the translational motion of water molecules is expected to be strongly suppressed like in solids. Thus, we name these confined supercooled water

molecules without translational freedom as quasi-2D confined amorphous ice.

To further explore this $\omega^3$ scaling found in the experimental data, we measured a series of samples with different levels of hydration. The results of the VDOS measurements on the various samples are shown in Fig. 3a, b. A lower hydration level corresponds to a smaller confinement size in the vertical $z$ direction, where the inter-layer distance inside the membrane can be varied from 7Å to a few nanometers when increasing $h$ from 0.1 to 0.7. We have fitted the experimental data with a single power-law function using as the upper value for the fitting window the point at which the VDOS of the bulk ice sample deviates from the Debye law and the shoulder of the higher energy vibrational peak appears, ≈4 meV. As shown in Fig. 3b–d, the power law of the low-energy VDOS changes gradually from 3 to 2 by varying $h = 0.1$ to 0.7. Such an outcome can be rationalized using the following logic. When the hydration level is low, the water molecules are mostly confined between the GOM layers and form a disordered solid phase; increasing the hydration level $h$, part of the water migrates away from being nano-confined between the layers to some large voids or even to the external surface of the GOM to form bulk crystalline ice, namely ice segregation[50,51]. This picture is supported by the small-angle X-ray scattering results (see Supplementary Fig. 1). As seen in

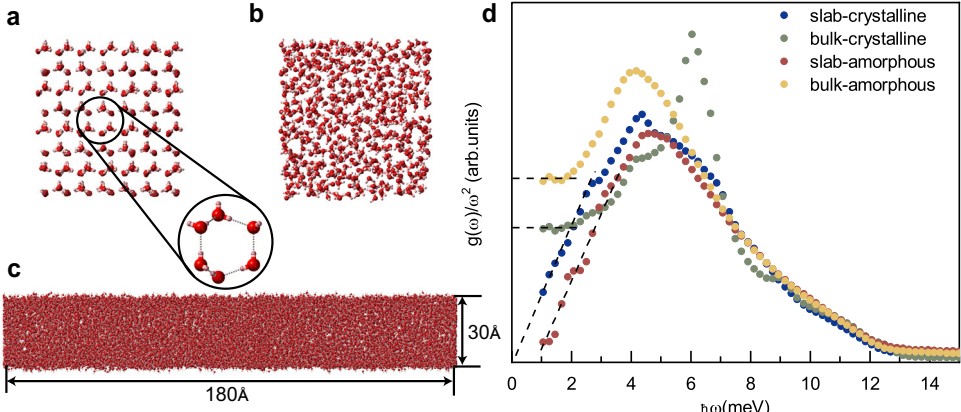

**Fig. 4 The molecular dynamics (MD) simulations performed at 120 K. a** The snapshot of the structure of the bulk ice sample. For visual purposes, only a few layers are kept. The zoom emphasizes its hexagonal structure. **b** The snapshot of the structure of the bulk amorphous ice sample. Also here, only a few layers are kept. **c** The amorphous ice slab sample with size 180 Å and confinement length along the $z$ direction 30 Å. Here, the original structure is kept. **d** The normalized VDOS in function of frequency obtained from the MD simulations for slab ice (blue), bulk crystalline ice (green), slab amorphous (red) and bulk amorphous ice (yellow). The horizontal dashed lines indicate the Debye level for the bulk systems. The diagonal dashed lines guide the eyes towards the $g(\omega) \sim \omega^3$ low-frequency scaling. The $y$-axes are presented in arbitrary units.

Fig. 3c, when the hydration is low, ($h = 0.1$), no Bragg peak is observed. However, the Bragg peaks appear when $h$ is above 0.4. We note that these Bragg peaks are rather narrow. This indicates that they result from bulk crystals instead of nanometer-sized crystals, where they would significantly broaden as seen in refs. [52,53]. This observation is further supported by the DSC measurement, where no first-order transition is observed at $h = 0.3$, but it appears at higher hydration levels, $h = 0.4$ or above. Moreover, as evident by SAXS data (see Supplementary Fig. 1), at high hydration levels (e.g. $h = 0.7$), the GOM interlayer-distance is suddenly reduced when cooling down to 253 K at which the Bragg peaks appear. Hence, at high hydration, e.g. $h = 0.4$ and $h = 0.7$, the crystalline ice is not formed confined between the layers, but it may exist within some large voids or on the external surface of the GOM, otherwise the inter-layer distance should increase rather than decrease as observed. Therefore, at any given hydration level, the system contains two separate components, nano-confined amorphous ice and large bulk ice crystals, which contribute with different scaling laws to the total density of states. The relative ratio between the two can be estimated using the DSC data (see Supplementary Note) and the results are presented in Fig. 3d. The nano-confined component obeys the $\omega^3$ scaling (assumed to be the same as the one discovered at $h = 0.1$) while the bulk crystal follows Debye's law $\omega^2$. Therefore, by fitting to a single power law, we obtain a smooth interpolation of the power from 3 to 2 by increasing $h$ from the strictly confined case towards the one dominated by the bulk crystal component (see Fig. 3b–d). The vanishing of the Debye contribution coming from the bulk crystal component is even clearer in the reduced VDOS presented in Fig. 3a. This agreement supports our previous argument that the density of states of the confined amorphous ice obeys a power-law $\sim \omega^3$. Such a cubic power law has also been recently reported in ref. [54] for a 2D model glass system. Despite the tempting similarities, our setup is different with respect to that of ref. [54] in various aspects. First, our is a quasi-2D system, rather than a small 2D system, with excitations also in the vertical $z$ direction. Second, as we will show with simulations in the next section, the $\omega^3$ power law is not an exclusive property of amorphous systems but it appears also in crystalline ones under strong confinement. All in all, the nature of the scaling discussed in this manuscript appears to be profoundly different with respect to the one reported in ref. [54].

Before moving on, let us comment on another interesting outcome of the experimental analysis. The bulk sample at 120 K displays a sharp resonance peak in the reduced VDOS at ≈7 meV (Fig. 3a). Despite the similarities with the boson peak anomaly in amorphous systems, most of the literature links its presence to a specific optical mode of the crystalline ice structure[55–57]. By decreasing the level of hydration, and therefore increasing the strength of confinement, we observe a shift of this peak towards lower frequencies and a broadening of its linewidth. This observation suggests that the effects of confinement decrease the lifetime of this mode and renormalize the energy of the optical modes similarly to what disorder or anharmonicity would do. Indeed, one could argue that confinement is itself a source of anharmonicity. Interestingly, the observed red-shift of the spectrum induced by the soft confinement is consistent with the results of ref. [42].

**Universality of the phenomenon checked with MD simulations at 120 K.** In order to confirm the universality of this power-law $\sim \omega^3$ for nano-confined solids and to prove that, as expected, it is not a peculiar property of the confined amorphous state, we perform all-atom molecular dynamics (MD) simulations on bulk ice, slab ice, supercooled water and slab supercooled water at 120 K, where the thickness of the slab along the $z$ direction is fixed to 3 nm. Supercooled water at such low temperature could be considered as amorphous ice. The different structure between the hexagonal crystalline ice and the amorphous ice is highlighted in panels (a) and (b) of Fig. 4. The geometry of the slab sample is shown in panel (c) of Fig. 4. The details of the simulations can be found in the Methods and Supplementary Note. The results of the simulations are shown in Fig. 4d. Both the amorphous and crystalline ices in the bulk phase show a very clear leveling-off in the reduced density of states at low frequency, signaling the presence of a strong Debye scaling law, $g(\omega) \sim \omega^2$. The Debye level is reasonably lower in the crystalline sample as compared to the bulk amorphous case because of a much larger value of the speed of sound. In the slab (confined) samples (Methods and Supplementary Note), the VDOS from the simulations shows clear differences in the low-frequency regime as compared to the bulk sample: the Debye leveling-off is absent and a $\omega^3$ scaling shows up, consistently with the neutron scattering experimental findings. In summary, the analysis of the MD simulations data confirms the outcomes of the

experiments and it also provides additional evidence for the universality of the phenomenon which does attain to any confined solid regardless of the crystalline or amorphous structure.

**Theoretical explanation and crossover frequency.** We provide a concise analytic derivation of the $\omega^3$ scaling observed for nano-confined solids both in the experimental data and MD simulations. Let us consider a three-dimensional system of linear size $L$. If the system were confined by atomically smooth and infinitely rigid boundaries (which is not the case for our experimental setup, see Fig. 2d), hard-wall boundary conditions (BCs) would apply, implying a net zero displacement of the systems' atoms/molecules near the boundary. In turn, the hard-wall BCs would lead to the usual "quantization" of the wavevector of the acoustic (elastic) waves that can propagate in the system: these are standing waves (eigenmodes of the wave equation) with $k = n\pi/L$, with $n$ being an integer (see Supplementary Note for more details).

In our system, however, the smooth hard-wall BCs do not apply because the confining boundaries are graphene oxide sheets, in which the basal plane is highly screwed and possesses different and spatially random-distributed oxide groups (hydroxyl, expoxy, and carboxy groups) (see the scanning electron microscope (SEM) image in Fig. 2d). An additional proof of this fact is provided in Supplementary Fig. 4 using the MD simulations. A crucial consequence of the lack of hard-wall BCs is the fact that the minimum length of wavevector is not $k = \pi/L$ and that the wavevector is not discrete but continuous. We have verified this statement using the numerical simulations. There (Supplementary Fig. 4 and Supplementary Note), it is found that the minimum wavevector is $\approx \frac{1}{4}\frac{\pi}{L}$, indeed smaller than $\pi/L$. No sign of the discreteness of the spectrum is observed either. Similar conclusions about the fundamental role of the boundary conditions in nano-confined systems have been reached in[35,42].

Going back to the main discussion, the number of states with wavenumber between $k$ and $k + dk$, i.e. in a spherical shell in $k$-space are given by

$$dn = \frac{1}{V_k} 4\pi k^2 dk. \tag{1}$$

where $V_k = (2\pi)^3/L^3$ is the $k$-state volume occupied by a single wavevector. Assuming a linear low-energy dispersion relation for the vibrational modes $\omega = vk$, which as we will see persists even at high level of confinement (see Supplementary Fig. 4 and Supplementary Note), one finally obtains the density of states as

$$g(\omega) = g(k)\frac{dk}{d\omega} = \frac{1}{V}\frac{dn}{dk}\frac{dk}{d\omega} \sim \omega^2, \tag{2}$$

where we omit the numerical prefactor since it is undetermined due to the non-smooth BCs (but importantly frequency independent). The $\omega^2$ scaling in Eq. (3) is the famous Debye's result[1,3]. Here, we have considered a single sound mode and we have neglected the existence of different polarizations as in realistic solids. Given the additive property of the density of states $g(\omega)$, one can easily consider this by simply summing the contributions from the different polarizations as

$$g(\omega) \approx \sum_p \omega^2 \tag{3}$$

where $p$ is the index labelling the polarizations. Importantly, the Debye derivation relies on an integration over all directions in the solid angle, reflected in the factor $4\pi$ in the above Eq. (1).

Let us now turn to a different situation in which one of the three bulk dimensions is confined but the atoms are still free to vibrate along it. This corresponds exactly to the setup considered in the experimental and simulation parts discussed above. We

follow closely the framework presented in[58], and we shall work without hard-wall BCs consistent with the discussion above. In particular, let us consider a geometry which is rotational invariant in the $(x, y)$ plane but in which the $z$ direction is confined in a range $z \in [0, L]$. We assume the size of the sample in the $(x, y)$ direction to be much larger than that in the $z$ direction, $L_z \ll L_x, L_y$. We use spherical coordinates, measuring the polar angle $\theta$ from the vertical $z$-axis. A pictorial representation of the geometry considered is shown in the panel a of Fig. 5. At a fixed angle $\theta$, the maximum wavelength allowed is $\lambda_{max} = L/\cos\theta$ because of the vertical confinement. Going to momentum space, this translates into a minimum wavevector allowed given by $2\pi\cos\theta/L$. This implies that below a certain crossover momentum $k_\times \equiv 2\pi/L$, the phase space is reduced by the effects of confinement. In particular, two spheres of radius $\pi/L$ centered at $(0, 0, \pm\pi/L)$ have to be excluded, as shown in panel b of Fig. 5. In the non-confined direction, i.e. for $\theta = \pi/2$, the macroscopic geometry does not provide any large-distance cutoff and the minimum value of $k$ is simply zero (ignoring the cutoff imposed by the size of the sample). Importantly, this minimum condition does not arise from the hard-wall BCs but simply from the geometry of the slab sample. The absence of hard-wall BCs is fundamental in allowing wavevectors below $k_\times$. Also, the effects of confinement are relevant only for wavevectors below the crossover scale $k_\times$. At higher wavevectors, the phase space is not affected since no minimum value nor angular dependence appears (see panel b of Fig. 5). This point will result important in the following.

As a result of this geometric constraint on $k$-space, below the crossover scale $k_\times$, the integration over the polar angles, which gave the $4\pi$ factor in Eq. (1), is now replaced by

$$2\int_{\cos^{-1}(Lk/2\pi)}^{\pi/2} \sin\theta d\theta \int_0^{2\pi} d\phi = 2Lk \tag{4}$$

where the complete integral in the $\phi$ angle reflects the $SO(2)$ rotational invariance in the $(x, y)$ plane. We note that the expression in Eq. (4) reduces to the standard Debye result when the wavevector approaches the crossover value $k_\times$. In particular, in that limit, the lower limit of integration tends to zero. Hence, we obtain that, below a certain threasold $k = k_\times$, the number of states with wavenumber in the range $\in [k, k + dk]$ is given by

$$dn \sim Lk^3 dk. \tag{5}$$

which clearly deviates from the Debye result. In Eq. (5), the numerical pre-factors have been neglected since they are not relevant for the present analysis. Following the same steps as before, one can deduce the VDOS at low frequency as

$$g(\omega) \sim \omega^3 \tag{6}$$

which is our main result of this section.

Equation (6) predicts a fundamental deviation from the Debye $\sim \omega^2$ law which arises because of the geometric constraints on the momentum phase space induced by spatial confinement in real space. This analysis is in agreement with the experimental and simulations outcomes of the previous sections and it is able to explain with a simple argument the $\omega^3$ universal scaling in solid systems under slab-confinement. Importantly, the same results would not be obtained for geometries with confinement along two spatial directions (cylinder) or three spatial directions (sphere) where, at least at low frequency, Debye's law is expected to work, as also confirmed in refs. [35,37–39,41,43].

Interestingly, our theoretical framework is also able to predict the crossover scale above which the Debye scaling re-appears. The latter is indeed given by $k_\times = 2\pi/L$, or alternatively, making use of the low-energy dispersion relation, by $\omega_\times \equiv 2\pi v/L$, with $v$ the

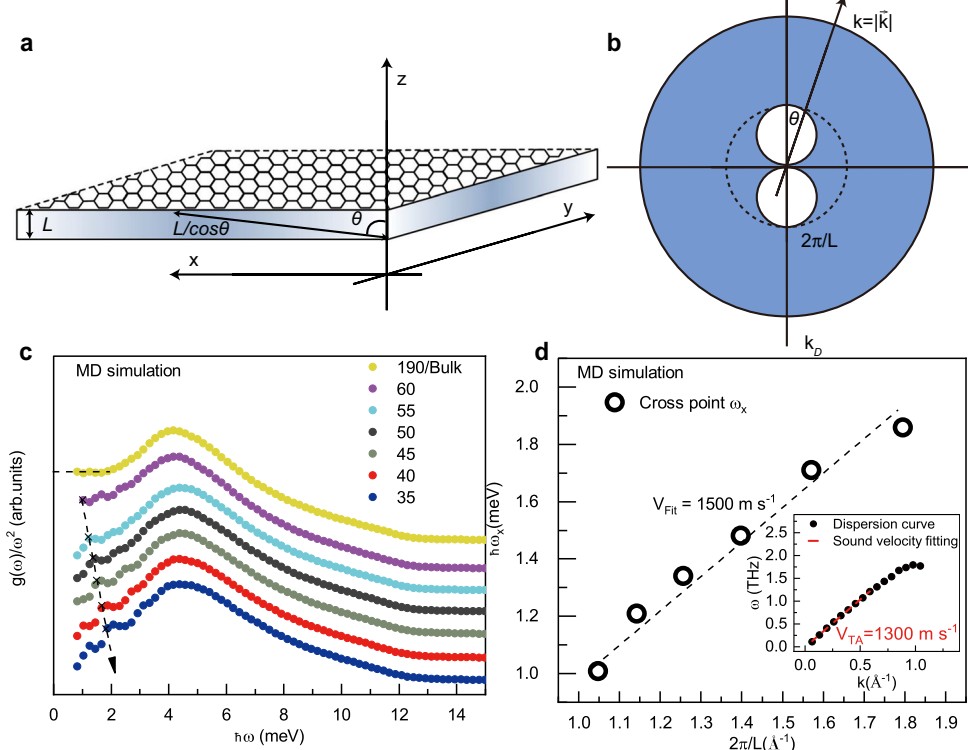

**Fig. 5 Theoretical model and validation from MD simulations. a** The confined geometry considered. The system size in the $(x, y)$ directions is much larger than that in the $z$ direction, $L_z \ll L_x, L_y$. The system is assumed to be rotational invariant in the 2-dimensional $(x, y)$ subspace. The angle $\theta$ is defined with respect to the vertical $z$ axes. **b** The wavevector phase space corresponding to the confined geometry of panel (**a**). **c** The normalized VDOS of slab amorphous ice with different confinement sizes $L$ (from the bulk sample in yellow color to the strongly confined sample in dark blue color). The black cross symbols show the crossover frequency $\omega_\times$ defined in the main text. **d** The crossover scale $\omega_\times$ as a function of the inverse confinement length $2\pi/L$. The dashed line is a linear fit whose result is reported in Eq. (7). The inset shows the dispersion curve of the transverse acoustic modes in amorphous ice and the fitting curve giving $v_{TA} \approx 1300\ \mathrm{m\ s^{-1}}$.

characteristic sound speed of the material. Because of the simplicity of the theoretical model, the crossover appears as a sharp transition at which the density of states is continuous but its derivative is not. Obviously, in more realistic situations, we do expect the crossover to be continuous and smoothed out by various effects including thermal fluctuations. In order to test this prediction of the theory, we have performed additional molecular dynamic simulations by dialing the size of the confined direction $z$ from very large (corresponding to a bulk system) to very small (corresponding to a nano-confined quasi-2D system). The results are presented in panel c of Fig. 5 by using a Debye reduced representation $g(\omega)/\omega^2$. In the bulk sample (yellow markers), a clear Debye level-off is visible at low frequencies. By decreasing the size of the confined region $L$, we observe the appearance of a low-frequency $\omega^3$ scaling as reported in the previous sections both in experiments and simulations. Importantly, using the data from simulations we are able to track the frequency at which the scaling of the density of state changes from the cubic scaling to the more standard Debye one (indicated with a × in panel c of Fig. 5). We then plot the position of this crossover scale as a function of the inverse confinement length $2\pi/L$. The data show a clear linear behaviour which is consistent with

$$\omega_\times = v_{\mathrm{fit}}\, 2\pi/L \quad \text{with} \quad v_{\mathrm{fit}} \approx 1500\ \mathrm{m\ s^{-1}} \qquad (7)$$

The fitted value for the sound velocity is in good quantitative agreement with the one extracted directly from the dispersion relation, $v_{TA} \approx 1300\ \mathrm{m\ s^{-1}}$ within a 14% error (see the inset of Fig. 5d). Here, the reference sound velocity (inset of Fig. 5d) is the transverse mode. If one used the average speed of transverse and

longitudinal modes, $3/\bar{v}^3 = 2/v_{TA}^3 + 1/v_{LA}^3$, with $v_{LA} \approx 3900\ \mathrm{m\ s^{-1}}$, an even better agreement with the value derived from Eq. (7) will be obtained, as $\bar{v} \approx 1480\ \mathrm{m\ s^{-1}}$.

In summary, the results from simulations confirm our theoretical framework and also prove that, despite the simplicity of the model, even its quantitative predictions are accurate.

## Discussion

In this work, we have reported the experimental observation of a low-frequency anomalous scaling in the vibrational density of states of nano-confined solids which violates the well-known Debye's law for bulk solid systems. In particular, using inelastic neutron scattering experiments on amorphous ice at 120 K nano-confined inside graphene oxide membranes, we have observed a low-frequency $\omega^3$ scaling law which substitutes the quadratic behaviour expected from Debye theory at low frequencies. This interesting experimental finding has been further confirmed by all-atom molecular dynamics simulations on confined ice in both crystalline and amorphous phases. Moreover, using a simple geometric analytical argument, a generalized law for the vibrational density of states of systems confined along one spatial direction has been derived. The appearance of this scaling is a consequence of the restricted wavevector phase space due to the geometric constraints imposed by confinement, while the nature of the low-energy vibrational modes does not change. Our picture is compatible with the idea that strong confinement produces a depletion of the low-energy part of the VDOS spectrum observed in several amorphous systems confined in nanopores[38,39,41,42,46], where nevertheless Debye's law is still obeyed at low frequencies.

Furthermore, our theory predicts that the Debye quadratic scaling re-appears above a characteristic frequency given by $\omega_\times = 2\pi v/L$, with $v$ being the speed of sound of the material. Using extensive molecular dynamic simulations, we have been able to confirm this prediction proving that our simple theoretical framework is not only able to explain the $\omega^3$ scaling but it also provides a good quantitative estimate of its frequency window. Finally, we stress that the nature of this scaling is not linked to the appearance of additional low-energy quasi-localized modes typical of amorphous systems as in[54] but it results from the geometric effects of confinement on the phase space of acoustic phonons.

Our analysis provides a universal answer to the fundamental question of the vibrational properties of quasi-2D nanoconfined solids and it paves a new path towards the understanding and study of the mechanical properties of condensed matter systems under confinement[59–61]. A direct consequence of this $\omega^3$ scaling in VDOS is to shift the acoustic modes towards the higher energies. Thus, the phonon-assisted transportation of energy, electron or proton in various electronic devices and biological systems of nano-meter confinement will be inevitably carried out more by the high-energy short-wavelength phonon modes. More specific functional changes of materials due to this new mechanism are left to be discovered. They may be significant to several fields including nano-mechanical systems, transport phenomena at the nano-scale and nano-scale manipulation of biological systems.

## Methods

**Sample preparation**. The GOM sample was synthesized using the modified Hummers' method[62]. The GOM sample was first dehydrated by heating it from room temperature to 313 K and then annealed at this temperature for 12 h in a vacuum to the dry condition. The oxidation rate of the GOM sample is 28%, which is determined by XPS. The dehydrated sample was sealed in a desiccator and exposed to the water vapor to allow water molecules to adsorb to the surface and the interlamination of the GOM sheets. The hydration levels were controlled by adjusting the expose time of the sample and the final values of hydration levels were determined by measuring the weight before and after the water adsorption.

**Differential scanning calorimetry (DSC)**. Differential scanning calorimetry (DSC) was used to measure the ratio between bulk ice and confined amorphous ice in GOM sample at low temperature. The DSC results of GOM at different hydration levels were performed by the DSC1 (METTLER TOLEDO). The samples were first annealed at 293 $K$ for 5 min, and then cooled down to 213 K at a cooling rate of 2 K/min to obtain the DSC data (see Supplementary Fig. 1). The DSC data were analyzed by the TA Trios software.

**Powder X-ray diffraction (PXRD)**. The powder X-ray diffraction data for GOM at different hydration levels were collected using a Rigaku Mini Flex600 X-ray diffractometer, with a Cu Kα source ($\lambda = 1.5406$ Å) operated at 40 kV and 15 mA at a scan rate of 10°/min from 10° to 60°. The PXRD data were analyzed by MDI Jade software.

**Small angle X-ray scattering (SAXS)**. SAXS characterizations were employed to monitor the interlayer distance evolution in GOM with temperature decreasing and ice freezing. The SAXS measurements were carried out at the BL16B1 beamline of the Shanghai Synchrotron Radiation Facility (SSRF). The wavelength of the X-ray was 1.24 Å. The SAXS patterns were collected by using a Pilatus 2M detector with a resolution of 1475 pixels × 1679 pixels and a pixel size of 172 $\mu m$ × 172 $\mu m$. The data acquisition time for SAXS was set as 10 s for each frame. The sample-to-detector distances of the SAXS is 258 mm, respectively.

**Scanning electron spectroscopy (SEM)**. SEM images were taken by a MIRA 3 FE-SEM with a 5 kV accelerating voltage.

**Neutron scattering**. The dynamic neutron scattering is described in terms of intermediate scattering function, $I(q, t)$ including incoherent and coherent terms $I_{inc}(q, t)$ and $I_{coh}(q, t)$ :

$$I_{inc}(q, t) = \sum_{j}^{N} b_{j,inc}^2 < \exp[-iq \cdot r_j(0)] \cdot \exp[iq \cdot r_j(t)] >, \quad (8)$$

$$I_{coh}(q, t) = \sum_{j}^{N} \sum_{i}^{N} b_{j,coh} b_{i,coh} < \exp[-iq \cdot r_i(0)] \cdot \exp[iq \cdot r_j(t)] >, \quad (9)$$

where $N$ is the total number of atoms, $b_{j,inc}$ and $b_{j,coh}$ are the incoherent and coherent scattering lengths of a given atom $j$, $r_j$ is the coordination vector of that atom, the bracket $<\cdots>$ denotes an ensemble and orientation average and $q$ is the scattering wavevector. $I_{inc}(q, t)$ contains the information about self-motions of atoms, and $I_{coh}(q, t)$ probes mostly interatomic motions. As the incoherent scattering cross section of hydrogen is at least one order of magnitude larger than incoherent and coherent scattering cross sections of other elements, the neutron signals collected on GOM hydrated in $H_2O$ are dominated by incoherent intermediate scattering function, $I_{inc}(q, t)$ and primarily reflect the self-motion of the water molecules. When $t = 0$, the coherent intermediate scattering function, $I_{coh}$, becomes the static structure factor, $I(q)$, characterizing the atomic structure of the system

$$I(q) = \sum_{j}^{N} \sum_{i}^{N} b_{j,coh} b_{i,coh} < \exp[-iq \cdot r_i(0)] \cdot \exp[iq \cdot r_j(0)] >. \quad (10)$$

The measured dynamic structure factor, $S(q, \omega)$ corresponds to the time Fourier transform of the intermediate scattering function,

$$S_{inc}(q, \omega) = \int_{-\infty}^{+\infty} I_{inc}(q, t) exp(i\omega t) dt, \quad (11)$$

$$S_{coh}(q, \omega) = \int_{-\infty}^{+\infty} I_{coh}(q, t) exp(i\omega t) dt, \quad (12)$$

where $\omega$ is the frequency and $\hbar\omega = \Delta E$ is the energy transfer between the incident and scattered neutrons. $S(q, \omega)$ provides information about the amplitude-weighted distribution of the dynamical modes in the sample with respect to frequency at any given wavevector $q$.

**Inelastic neutron scattering (INS)**. As the incoherent scattering cross-section of hydrogen is at least 1 order of magnitude larger than incoherent and coherent scattering cross-sections of other elements, the neutron signals are dominated by incoherent scattering function, and primarily reflect the self-motion of the water molecules. The experimental vibrational density of states (DOS) $g(\omega)$ can be obtained from the dynamic structure factor, $S(q, \omega)$ using the function[63]:

$$g(\omega) = \int \frac{\hbar\omega}{q^2} S(q, \omega)(1 - e^{-\frac{\hbar\omega}{k_B T}}) dq \quad (13)$$

where $\hbar$ is the Planck constant, $q$ is the scattering wavevector, $\omega$ is the frequency related to the energy transfer, $k_B$ is the Boltzmann constant, and $T$ is the temperature. The experiments for samples of pure water, GOM absorbed by $H_2O$ with $h$ (gram water/gram GOM) of 0.1, 0.4 and 0.7 were conducted by using a time-of-flight (TOF) cold neutron polarization analysis spectrometer PELICAN at ANSTO in Australia with an energy resolution $\Delta E = 0.35$ meV (upwards to 1 meV or so[49]) and the energy ranges up to 24.8 meV with the energy gain mode used in the $q$ range from 0.08 Å$^{-1}$ to 4.5 Å$^{-1}$. The incident energy is 14.9 meV with the wavelength of 2.345 Å. The samples were contained inside aluminum foils in a solid form and sealed in aluminum sample cans in a helium atmosphere. The empty can signal was subtracted at each temperature. The detector efficiency in the data was normalized using a vanadium standard. All steps were performed with standard routines within the LAMP software package[64] and the scripts are available upon request. The experiment for pure dry GOM was conducted by using a high-intensity Fermi-chopper spectrometer 4SEASONS at J-PARC in Japan[65]. The measurement was done with multi-incident energies[66] and the data with incident energy 27.1 meV was chosen to cover the energy range up to 17.2 meV and the $q$ range from 0.225 Å$^{-1}$ to 7 Å$^{-1}$. The energy resolution at the elastic line is $\Delta E = 0.8$ meV. Similar processing steps to other samples data above were performed on Utsusemi[67] and Mslice software packages.

**Molecular dynamics (MD) simulations**. The MD simulations were performed using LAMMPS[68] to simulate ice and supercooled water at 120 K. The supercooled water is simply obtained by simulating bulk liquid water at room temperature and then cooling it down to 120 K. A temperature of 120 K is low enough to freeze out the translational degrees of freedom; therefore, one can consider the disorderly packed water at such temperature as amorphous ice. The equilibration of the MD systems was performed in constant temperature and constant pressure ensemble, using the Nosé-Hoover thermostat and Parrinello–Rahman barostat to control the temperature and pressure, and then switched to NVT ensemble to calculate dynamical properties. The timestep is set as 2 fs. The inter-molecule potential of $H_2O$ used is TIP4P/2005[69], which shows good accuracy for ice and supercooled water[70]. To reduce the finite-size effect, the simulation are performed using 360,000 and 216,000 molecules for the ordered and disordered structures respectively, with the configuration edge sizes of 180 and 200 Å. Both crystal and amorphous structures were equilibrated at 120 K at 1 atm. The slab structures were then cut from the bulk system to a thickness of 30 Å. We freeze the bottom and top layer (~3 Å) of the slab systems to force the structure to remain 2D confined during the whole simulation. In order to find the optimal volume of the ice slab, we performed a simulation for ice slab in the NPT ensemble. The position of the top and bottom layers are rescaled to new positions when the simulation box changes. A snapshot of the slab geometry can be found in Fig. 4c in the main text.

The VDOS is calculated by the Fourier transform of the oxygen velocity autocorrelation function:

$$C_v(\omega) = \int_{-\infty}^{\infty} C_v(t) \exp(-i\omega t) dt. \qquad (14)$$

The velocity auto-correlation function (VAF) is defined as

$$C_v(t) = <\mathbf{v}(0) \cdot \mathbf{v}(t)> \qquad (15)$$

where $\mathbf{v}(0)$ are the oxygen velocities.

## Data availability

The datasets generated and analysed during the current study are available upon reasonable request by contacting the corresponding authors.

## Code availability

The code that supports the findings of this study is available upon reasonable request by contacting the corresponding authors.

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

## Acknowledgements

The authors thank Reiner Zorn, Lijin Wang, and Jie Zhang for useful discussions. The authors thank Dr. Xiaran Miao from BL16B1 beamline of Shanghai Synchrotron Radiation Facility (SSRF) for help with synchrotron X-ray measurements. We also appreciate the assistance from the Instrumental Analysis Center of Shanghai Jiao Tong University for SEM, PXRD, DSC measurements. M.B. acknowledges the support of the Shanghai Municipal Science and Technology Major Project (Grant No.2019SHZDZX01). C.Y. acknowledges the support of the NSF China (11904224). This work was supported by NSF China (11504231, 31630002, and 22063007), the Innovation Program of Shanghai Municipal Education Commission and the FJIRS-M&IUE Joint Research Fund (No. RHZX-2019-002). The neutron experiment at the Materials and Life Science Experimental Facility of the J-PARC was performed under a user program (Proposal No. 2020I0001). The beam time supported by ANSTO through the proposal number P7273.

## Author contributions

Y.Y. performed the experimental measurements; M.B. and L.H. conceived the idea of this work, C.Y. implemented the MD simulations, L.Z. made the experimental sample, A.E.P., M.B., and A.Z. developed the theoretical model; R.K., M.N., and D.Y. helped with the inelastic neutron scattering experiments in the Japan and Australia, respectively; Y.Y. and C.Y. contributed equally to this work. M.B. and L.H. jointly supervised this work. All the authors contributed to the writing of the manuscript.

## Competing interests

The authors declare no competing interests.
