## [Peer Review File · Nature Communications]

The ω^3 scaling of the vibrational density of states in quasi-2D nanoconfined solidsREVIEWER COMMENTS

Reviewer #1 (Remarks to the Author):

The manuscript by Yu et al. employs a series of inelastic neutron scattering and molecular dynamics simulations to study the scaling behavior in amorphous solids under nanoconfinements at low frequency. The authors find a clear cubic scaling of the VDOS at low frequencies, as opposed to Debye's scaling of ω^2 . The experiments are carefully performed by sandwiching the samples in graphene oxide membranes, which can provide soft confinements in a controlled manner. The authors also provide a theoretical ration to observed scaling with regards to geometric considerations of the system. While I believe in general the work warrants publication in Nat. Comm., some clarifications and additional information may be required before publication:

1- MD simulations have been performed, which in general of great in boosting one's confidence in observed scaling in neutron scattering. However, the results are very briefly discussed, with a single graph presented and some snapshots of the bulk ice, as opposed to confined geometry. Having access to all spatiotemporal configurations in the simulations, presenting the actual structure can be quite informative. Also, some anomalous behavior is observed in Fig. 4D for the bulk ice, which should be clearly described.

2 - The experimental VDOS measurements in Fig. 3B show a transition from scaling of 2 to 3, rather gradually while changing the confinement layer; nonetheless, I'm not sure that I agree with the slopes presented on the graph, specially with GOM 0.4, the slope is very similar to 3. If so, some discussion will be necessary to clarify the transition from 2 to 3 scaling. Along the same line, Fig. 3 D for the hydration level shows 6 different samples (data points), but all results are presented for 4 samples. Some clarification would be useful.

3- Details of matching the mechanics of the GOM in MD simulations with the experiments will be informative.

4- Minor comment on some grammatical error throughout; but this should not affect the overall assessment of the manuscript as I'm confident all those will be carefully looked at prior to publication.

Considering the comments above, I believe that the work contributes fundamentally to a wide range of areas, and the results presented are synergistically confirming the hypotheses put forward by the authors. As such, I am happy to recommend publication upon addressing the comments above

Reviewer #2 (Remarks to the Author):

This paper reports the scaling of vibrational density of states in a quasi-2D confined solids. The sample was amorphous ice confined by graphene oxide membranes (GOMs). The confinement size L is controlled by hydration level. Inelastic neutron scattering measurements were employed to determine the vibrational density of states (DOS). The authors found that DOS scales with ω^3 when L is small, and gradually approaches ω^2 when L increases. Molecular Dynamics simulations were carried out to confirm the findings from inelastic neutron scattering experiments. Finally, a theoretical analysis was carried out with the aim to demonstrate why the DOS behaves in such a way in a confined system.

Phonon DOS in glasses has long attracted a lot of interests. In spite of extensive studies, however, there are still wide-range of disagreements between experiments and simulations, and also amongst different simulation studies themselves. In this regard, the present study, which was carefully orchestrated, is interesting and could make an important contribution to our understanding of the phonon dynamics in glass.

However, I have some concerns which I would like the authors to address.

1. A numerical study was recently published in Phys. Rev. Lett. [PRL, 127, 248001 (2021)] on low-frequency excess vibrational modes in a 2D model glass, where the authors found that the DOS scales

with w^2 in a typical 2D glass but the scaling changes to w^3 for a small system. I would like the authors to consider this paper and address how their findings reconcile or differ from this paper.

2. The authors attempted a theoretical analysis to show that there are extra modes at low energies due to the peculiarities of the confinement. The argument hinges on Eq. (4), but I cannot see how this was derived. The authors referred to ref. [79], but this is an unpublished result posted online.

Furthermore, I had trouble to access the file (invalid link). Given that Nature Communications does not impose a page limit, I suggest that the authors elaborate on the development of Eq. (4) and incorporate the key parts of ref. 79 in Supplemental Materials.

3. Along the same line, please also show analytically how the scaling transitions from w^3 to w^2 as L increases.

4. Can the simulation also confirm a transition from w^3 to w^2 as L increases?

5. The authors discussed the data in the low-energy regime leading up to the Boson peak. What is the energy of the Boson peak in these materials?

Reviewer #3 (Remarks to the Author):

This is a nice combination of experimental and theoretical work. The topic is of general interest. Therefore, the manuscript should be published. Nature Communication seems to be a good place for that.

In the abstract it is written that the vibrational density of states and the specific heat is well described by the Debye theory. This is only true for crystalline materials where a huge variety of materials like glasses deviations are found. The authors should be more specific with that. The same is true for the first part of the introduction. The Debye assumption is fulfilled at low enough frequencies seems to be trivial for bulk systems because the dispersion relation could be always approximated by a linear law.

I fully agree with the authors that in confined and/or fractal systems a linear approximation is not appropriate because some states cannot be realized and reach under such circumstance. The presented experimental data seems to be nice examples for that. Nevertheless, other numerical approaches to the low frequency density of states show that also under confinement situation becomes quadratic in the frequency dependence. See Phys. Rev. B: Condens. Matter Mater. Phys., 2010, 81, 054208. At least this paper should be cited.

In figures 2,3 and 4 at the x-axis ω is used. General in physics ω is used as a symbol for the frequency (see also equ. 2). In these figures energy values are given. This is not consistent.

Minor comment:

In figure 7 it should be THz.

Temperature differences should be given in K not in C in accordance with international regulations.

Response to the Referees for manuscript NCOMMS-21-37774-T

Dear Referees,

thanks a lot for reviewing our manuscript and for the positive assessment of our work. In the following, we reply point by point to the comments of all the Referees. We have thoroughly revised and improved the manuscript (see below for details). The changes in the manuscript are left in **red color** to facilitate the job of the Referees.

Moreover, inspired by the comments of the Referees, we have added additional material in the manuscript. In particular, (I) we have added the analytic prediction from theory of the crossover frequency from the ω^3 scaling to the Debye ω^2 scaling and (II) we have performed a further analysis of the DOS in the simulations for different confinement sizes and confirmed the theoretical prediction in point (I).

The point-by-point response to the Referees' questions are provided below with the original Referees' questions marked in blue Italic. We hope the Referees and the Editor will find now our work suitable for publication in Nature Communications.

Referee 1

The manuscript by Yu et al. employs a series of inelastic neutron scattering and molecular dynamics simulations to study the scaling behavior in amorphous solids under nanoconfinements at low frequency. The authors find a clear cubic scaling of the VDOS at low frequencies, as opposed to Debye's scaling of ω^2 . The experiments are carefully performed by sandwiching the samples in graphene oxide membranes, which can provide soft confinements in a controlled manner. The authors also provide a theoretical ration to observed scaling with regards to geometric considerations of the system. While I believe in general the work warrants publication in Nat. Comm., some clarifications and additional information may be required before publication:

We thank Referee 1 for his/her positive evaluation of our work. In the following we address all the comments of Referee 1 point-by-point.

MD simulations have been performed, which in general of great in boosting one's confidence in observed scaling in neutron scattering. However, the results are very briefly discussed, with a single graph presented and some snapshots of the bulk ice, as opposed to confined geometry. Having access to all spatiotemporal configurations in the simulations, presenting the actual structure can be quite informative. Also, some anomalous behavior is observed in Fig. 4D for the bulk ice, which should be clearly described.

We thank Referee 1 for the comments and suggestions.

Additional details about the MD simulations, the systems used and the corresponding structures have been added in the Supplementary Material (Fig.R-1(a)-(f) in this Reply). The pair distribution function analysis in the radial direction, revealing additional information about the spatial structure, has been added as well to confirm the difference between the crystalline and amorphous setups.

In Fig.4D of the manuscript, a sharp peak in the VDOS of bulk ice is observed around ≈ 1.5 THz (6.2 meV). In order to investigate its origin further, in Fig.R-1(g),(h), we have computed the velocity auto-correlation function in different coaxial directions to obtain the vibrational density of states in the various directions. From there, we can conclude that such a sharp peak comes from the vibration perpendicular to the [0001] basal plane of the hexagonal ice. In both the amorphous and slab-crystalline samples, this peak is broadening and moves slightly to lower frequencies, ≈ 1.1 Thz (4.5 meV). Such effect can be explained by an increase of the linewidth corresponding to this mode. In the case of the amorphous systems, this broadening of the linewidth naturally arises because of structural disorder. In the case of the slab-crystalline sample, the spatial confinement on the z direction reduces the constructive interference and thus decreases the peak intensity. Regarding the nature of this peak, there is yet no consensus in the literature. Some works identify this excess mode as a genuine boson peak (PhysRevLett.85.3185; Physica B: Condensed Matter, 316, 493-496.), others (PhysRevLett.85.4100; PhysRevB.78.064303,PhysRevB.77.104306) attribute this excess to a specific optical mode of the ice structure. Despite being an interesting question, it is only tangential to the scope of the present work; therefore, we are not in the position of saying more about this peak. We have added the above discussion and Fig.R-1(g),(h) in the revised Supplementary Material to clarify this point.

Figure R-1: Snapshots of the crystalline and amorphous ice structures used in the simulations from different perspectives. (a) Top view of crystalline ice. (b) Side view of crystalline ice. (c) Top view of amorphous ice. (d) Side view of amorphous ice. (e) The radial distribution function of bulk ice. (f) The radial distribution function of slab ice. (g) The VDOS of bulk ice calculated from the velocity auto-correlation function along different axes. (h) The VDOS of slab ice calculated from the velocity auto-correlation function along different axes.

The experimental VDOS measurements in Fig. 3B show a transition from scaling of 2 to 3, rather gradually while changing the confinement layer; nonetheless, I'm not sure that I agree with the slopes presented on the graph, specially with GOM 0.4, the slope is very similar to 3. If so, some discussion will be necessary to clarify the transition from 2 to 3 scaling.

Following the suggestion of the Referee, we have revisited and improved the fitting. Additionally, the details of the fitting procedure were not totally clear in the previous version of manuscript, but were now clarified in the revised one. Briefly, the power law reported is obtained by choosing a proper energy range for the fitting. More precisely, we have fixed this range by referencing the results in bulk ice, which were Fig.3(a) of the original manuscript and presented in Fig.R-2(a) of this reply. There, one can notice that the Debye scaling law is robust only up to ≈ 4 meV (indicated by a vertical dashed line in Fig.R-2(a)), as the shoulder of the strong vibrational peak discussed before enters at higher energy. Therefore, this value is taken as the upper bound for the energy window of the fit. For consistency, all the data are fitted in the range of 2.4meV to 4meV, where the lower bound is set as 2.4 meV. Using this more conservative procedure, the power law of the sample at $h = 0.4$ is around 2.6 (see Fig.R-2(b)) and indeed lower than that of the sample at $h=0.1$, which is 3. As shown in Fig.R-2(c), within the experimental error the power law fittings well modeled the experimental data. The previous value for $h = 0.4$ was overestimating the power because higher energy points were used in the fitting and the shoulder of the strong vibrational peak was affecting the final result. We have improved the fitting procedure and added in the text more details about how it is performed. Additionally, the corresponding figure (Fig.3(b),(d)) in the manuscript has been modified.

The continuous transition between the ω^2 to ω^3 scaling in the experimental setup can be understood as follows. When the hydration level is low, the water molecules are mostly confined between the GOM layers and form confined amorphous ice, showing the ω^3 scaling derived in the manuscript. By increasing the hydration level, part of the water molecules at low temperature moves to some large voids or even to the external surface of the GOM to form bulk crystalline ice, contributing to the total DOS with the standard ω^2 Debye scaling. Therefore, at any given hydration level, the system contains two separate components, a 2D-confined disordered phase and 3D bulk ice, which contribute with different scaling laws to the total

density of states. As a result of summing these two terms, and using a single power-law fitting function, one obtains a continuous crossover between the two powers which is mainly the manifestation of how the corresponding prefactors change upon hydration. The relative mass ratio between the two components can be roughly determined by using Differential scanning calorimetry. As shown in Fig.R-2(d), the mass proportion of bulk ice gradually increases from 0 at $h=0.1$ to $\approx 67\%$ at $h=1.2$, implying that the higher the hydration level the less the confined disordered phase and the more the bulk crystalline ice. Because of this competition, and the unavoidable bulk ice component in the experimental setup, a single effective power law fitting results into a graduate change of the scaling from 3 to 2 upon increasing h . We have expanded and improved the discussion about this point in the revised manuscript.

Along the same line, Fig. 3 D for the hydration level shows 6 different samples (data points), but all results are presented for 4 samples. Some clarification would be useful.

The discrepancy on the number of data points between neutron and DSC in Fig.3D of the main text results from the different cost of these two methods. The beamtime of the inelastic neutron scattering is limited, and we collected data for 9 hours per sample at each temperature to get sufficient statistics. The DSC measurement was conducted after the neutron scattering, and it was performed by using a household instrument for 1.5 hour per sample per temperature. Therefore, we were able to measure more samples covering the range of hydration levels studied by neutron scattering. We have clarified this difference in main text.

Figure R-2: (a) The Debye normalized VDOS $g(\omega)/\omega^2$ for bulk crystalline ice measured in this work at 120K. The vertical dash lines indicate the range of the energy window (2.4 meV to 4 meV) used for fitting. (b) The log-log scale plot of the VDOS for bulk ice, GOM sample at $h=0.7, 0.4, 0.1$. The dashed lines represent the results of our fit. (c) The zoom view diagram of fitting interval in (b). (d) The fitting power of the low frequency VDOS data for different hydration levels (red). The dashed lines indicate the ω^3 scaling and the standard quadratic Debye's law. The DSC data showing the mass proportion of bulk ice for the corresponding hydration levels (blue).

Details of matching the mechanics of the GOM in MD simulations with the experiments will be informative.

We have performed additional MD simulations on GOM-sandwiched water (see the snapshot in Fig.R-3(b)). The obtained DOS is in qualitative agree-

ment with that from the slab setup without GOM. In the main text, we still use the slab simulation without GOM for our analysis and discussion. The reasons are the following: first, the slab simulation is easy to set up for both crystalline and amorphous phases, and thus one can make a fair comparison of the two systems under the same confinement. In contrast, GOM has a rather rough and curvy surface (see Fig.R-3(b) below), and thus the simulation for confined crystalline ice becomes difficult. Secondly, precise control of the thickness of the sample is needed in order to test the theoretical derivation for the crossover scale between the ω^3 and ω^2 scalings (see Fig.R-4 and the related reply to Referee 2 for more details about the crossover scale). Again, the rough and curvy surface of GOM prevents such an analysis. In order to explain in more detail this point, we have added the above discussion and Fig.R-3 in the revised Supplementary Information to show that the scaling law observed experimentally can also be seen in the MD simulation of GOM-sandwiched water.

Figure R-3: **(a)** The normalized VDOS in function of frequency obtained from the MD simulations for slab crystalline ice, bulk crystalline ice, slab amorphous ice, bulk amorphous ice and amorphous ice confined in GOM. **(b)** The actual structure of GOM confined amorphous ice in simulations. The epoxy group on the GOM is marked by a green oxygen atom and the hydroxyl group by a blue one.

Minor comment on some grammatical error throughout; but this should not affect the overall assessment of the manuscript as I'm confident all those will be carefully looked at prior to publication.

We have revised carefully the text and fixed the spotted grammatical errors. Thanks for pointing this out.

Considering the comments above, I believe that the work contributes fundamentally to a wide range of areas, and the results presented are synergistically confirming the hypotheses put forward by the authors. As such, I am happy to recommend publication upon addressing the comments above

We would like to thank Referee 1 once again for the helpful comments and for the positive evaluation of our work. We have carefully considered the points above and improved our manuscript accordingly. We hope that Referee 1 will find it now suitable for publication in Nature Communications.

Referee 2

This paper reports the scaling of vibrational density of states in a quasi-2D confined solids. The sample was amorphous ice confined by graphene oxide membranes (GOMs). The confinement size L is controlled by hydration level. Inelastic neutron scattering measurements were employed to determine the vibrational density of states (DOS). The authors found that DOS scales with ω^3 when L is small, and gradually approaches ω^2 when L increases. Molecular Dynamics simulations were carried out to confirm the findings from inelastic neutron scattering experiments. Finally, a theoretical analysis was carried out with the aim to demonstrate why the DOS behaves in such a way in a confined system. Phonon DOS in glasses has long attracted a lot of interests. In spite of extensive studies, however, there are still wide-range of disagreements between experiments and simulations, and also amongst different simulation studies themselves. In this regard, the present study, which was carefully orchestrated, is interesting and could make an important contribution to our understanding of the phonon dynamics in glass.

We thank Referee 2 for the positive evaluation of our work and for finding our study interesting. Some of his/her questions prompted us to improve the theoretical analysis and to add new simulations in support of it.

However, I have some concerns which I would like the authors to address.
1. A numerical study was recently published in Phys. Rev. Lett. [PRL, 127, 248001 (2021)] on low-frequency excess vibrational modes in a 2D model

glass, where the authors found that the DOS scales with ω^2 in a typical 2D glass but the scaling changes to ω^3 for a small system. I would like the authors to consider this paper and address how their findings reconcile or differ from this paper.

We thank Referee 2 for pointing us to this recent interesting paper. In the PRL paper mentioned [PRL, 127, 248001 (2021)], the author found that the low-frequency excess vibrational modes, obtained by subtracting the Debye contribution from the overall density of states, in a 2D model glass scales with ω^2 , but it changes to ω^3 for a small system. Despite the apparent similarities, our system is quite different from that discussed in the reference mentioned above. The authors of the PRL consider a purely 2D system without a third direction. Our system is not a 2D system, but rather a 3D system with the third dimension strongly confined. Importantly, there is vibrational dynamics also along the third confined dimension, which is the key for the ω^3 scaling observed here.

In addition, using simulations, we are able to show that this ω^3 scaling is not an exclusive feature of amorphous systems but it appears also in ordered structures under confinement, while the PRL studies glassy systems only. Moreover, the density of states studied in the present work is the total one obtained without subtracting the Debye contribution. Finally, our physical interpretation of the ω^3 scaling is quite different since it does not involve any low-frequency excess vibrational modes.

In Summary, at this point we believe that our scaling is not of the same nature of the one reported in the PRL mentioned. In the revised manuscript, we have cited the PRL paper and added the above discussion.

The authors attempted a theoretical analysis to show that there are extra modes at low energies due to the peculiarities of the confinement. The argument hinges on Eq. (4), but I cannot see how this was derived. The authors referred to ref. [79], but this is an unpublished result posted online. Furthermore, I had trouble to access the file (invalid link). Given that Nature Communications does not impose a page limit, I suggest that the authors elaborate on the development of Eq. (4) and incorporate the key parts of ref. 79 in Supplemental Material.

Let us clarify our interpretation further. We do not attribute the appearance of this novel scaling to extra modes at low energies, as usually done in the context of amorphous systems. On the contrary, we show that the geometric confinement, together with the absence of periodic boundary conditions,

modifies, by itself, the Debye contribution (below a certain frequency scale, see more below) changing the scaling from quadratic to cubic at low energy. "Eq.(4)" mentioned by the referee is now published in Phys. Rev. Materials 5, 035602 and it has been updated in the manuscript. Nevertheless, for the sake of clarity, we are happy to provide a few more details of the derivation in the manuscript. We have expanded the theoretical derivation with more details in the revised manuscript and updated the citation with the published version.

Along the same line, please also show analytically how the scaling transitions from ω^3 to ω^2 as L increases. Can the simulation also confirm a transition from ω^3 to ω^2 as L increases?

We would like to thank Referee 2 for these two questions since they urged us to greatly improve the exposition of the theory part and to add additional simulations providing further and clear evidence for its validity.

Let us start by explaining in more detail the theoretical framework. We consider a cylindrical system confined to length L in the z direction and for simplicity we assume the other directions (x, y) to be infinitely extended.¹ We use spherical polar coordinates, measuring the polar angle θ from the z axis (see Fig.R-4(a)). If measured at an angle θ from the z confinement axis, the extent of the confined medium is $L/\cos\theta$ (Fig.R-4(a)). Taking this value to be the maximum allowed wavelength in that direction, and using the absence of hard-wall boundary conditions (which is proved with simulations in the Supplementary Material), we obtain $k_{\max} = 2\pi \cos\theta/L$. In the range $0 \leq \theta \leq \pi$, this equation describes two spheres with radius π/L , centred at $(0, 0, \pm\pi/L)$ in k -space as shown in Fig. R-4(b). This implies that the phase space of allowed momenta $k \equiv \vec{k}$ becomes angle dependent below $k_{\times} = 2\pi/L$. In particular, momenta lying within the two spheres, white region in Fig. R-4(b), are not allowed. Notice that the geometric confinement has no effects for $k > k_{\times}$ and that $k < k_{\times}$ would not be allowed in presence of hard-wall boundary conditions.

By now inverting the constraint on k_{\max} in terms of a minimal angle $\theta_{\min} = \arccos(k/k_{\times})$, we arrive at Eq.4 in the main text from which we derived the novel ω^3 scaling, after converting the results from momentum k to frequency

¹From a physical perspective, this is tantamount to say that their extension is much larger than L . The finite size corrections can be directly computed (see Phys. Rev. Materials 5, 035602) but they are not relevant for our purpose.

Figure R-4: **(a)** The geometry of our system. **(b)** The effects of confinement on the wavevector phase space. The allowed region is displayed in blue color. The absence of hard-wall bcs allows for wavevectors with $k < 2\pi/L$ (dashed black line). The 1D confinement then imposes ulterior restrictions below such a threshold which are shown as white regions. **(c)** The Debye normalized DOS in function of the confinement scale L from 190 (considered as the bulk sample) to 35. The \times indicates the crossover frequency and the dashed arrow guides the eyes of the Reader towards the motion of ω_\times . **(d)** The dependence of the crossover frequency with respect to the inverse confinement scale $2\pi/L$. The dashed line is a fit giving $\omega_\times = v_{fit} 2\pi/L$ with $v_{fit} \approx 1500$ m/s. The inset shows the dispersion curve of the transverse acoustic modes in amorphous ice and the fitting curve giving $v_{TA} \approx 1300$ m/s.

ω , using the dispersion relation $\omega = vk$. We emphasize that the novel scaling is expected to roughly appear at frequencies below $\omega_\times = vk_\times$ with v being the characteristic speed of sound of the material. In summary, our

theoretical model predicts that:

- Above $\omega_{\times} = 2\pi v/L$, there is no effect of the geometric confinement and the density of states of the solid system is expected to follow the standard Debye law $g(\omega) \sim \omega^2$.
- Below ω_{\times} , a novel ω^3 scaling appears because of the geometrical effects of confinement on the phase space of the low energy vibrational modes.

From a theory point of view, this crossover is sharp, meaning the density of states is continuous at ω_{\times} but its derivative is not. Obviously, this feature is a result of the simplifications required to keep the theoretical model analytically tractable, and in the realistic situation it will be smeared out by several effects, including thermal fluctuations.

In order to answer the second question of Referee 2, and test the validity of the predictions from our theory, we have performed more simulations by varying the confinement scale L from small (nanoconfined system) to large (bulk material). The results are presented using the Debye reduced density of states $g(\omega)/\omega^2$ in Fig.R-4(c). The bulk sample (yellow points therein) shows a clear plateau at low frequencies confirming the validity of Debye law, $g(\omega) \sim \omega^2$. By decreasing the confinement scale L , we notice a gradual appearance of a novel ω^3 regime at low frequencies, up to a crossover scale indicated with \times in the plot. The crossover scale ω_{\times} moves to larger frequencies by reducing L , as expected from the theoretical considerations described above.

In order to quantitatively validate our theory, we have tracked the position of the crossover frequency ω_{\times} as a function of the inverse confinement scale $2\pi/L$. In Fig.R-4(b), we report a linear scaling between the two quantities, confirming the theory predictions. A numerical fitting gives a value:

$$v_{fit} = \frac{\omega_{\times} L}{2\pi} \approx 1500\text{m/s} \quad (\text{R-1})$$

which is in good quantitative agreement with the actual value of sound velocity extracted from the dispersion relation, $v \approx 1300$ m/s within a 14% error(see the inset of Fig.R-4d). Here, the reference sound velocity (inset of Fig.R-4d) is the transverse mode. If one used the average speed of transverse and longitudinal modes, $3/\bar{v}^3 = 2/v_{TA}^3 + 1/v_{LA}^3$, with $v_{LA} \approx 3900$ m/s, an even better agreement with the value dervied from Eq.(R-1) will be obtained, as $\bar{v} \approx 1480$ m/s.

In summary, the new simulations analysis confirms the validity of our theoretical framework not only in predicting the novel low frequency scaling $\sim \omega^3$ under confinement but also in estimating the crossover frequency to the Debye scaling.

We have re-written the theory part, added the new results from the simulations together with a lengthy discussion regarding the confirmation of the validity of the theory and its predictions. Fig.R-4(b)-(d) have also been added to the revised text (Fig.5(b),(c) and (d)).

The authors discussed the data in the low-energy regime leading up to the Boson peak. What is the energy of the Boson peak in these materials?

Both low density amorphous ice (LDA) and high density amorphous ice (HDA), whose VDOS is plotted in Fig.2(a) of the main text, present a clear peak around $\approx 5 - 6$ meV. This peak in amorphous ice has been reported in several instances, see for example (Phys. Rev. Lett. 94, 125506) from which our reference data are taken. Nevertheless, there is yet no consensus whether such a peak is a genuine boson peak as in other glassy materials or rather arises because of the presence of a optical-mode resonance specific of the water structure, [(PhysRevLett.85.4100; PhysRevB.78.064303; PhysRevB.77.104306)]. Given the fact that the nature of that peak is only tangential to the scope of the present work, we are not in the position of making any claim regarding this peak. We have clarified this point further in the revised manuscript.

Referee 3

This is a nice combination of experimental and theoretical work. The topic is of general interest. Therefore, the manuscript should be published. Nature Commication seems to be a good place for that.

We thank Referee 3 for his/her positive opinion about our work and for suggesting its publication in Nature Communications.

In the abstract it is written that the vibrational density of states and the specific heat is well described by the Debye theory. This is only true for crystalline materials where a huge variety of materials like glasses devia-

tions are found. The authors should be more specific with that. The same is true for the first part of the introduction. The Debye assumption is fulfilled at low enough frequencies seems to be trivial for bulk systems because the dispersion relation could be always approximated by a linear law.

The Referee is certainly right. Our presentation in the abstract and in the introduction was sloppy. The Debye law does not apply to amorphous materials in which important deviations are found. Following the suggestion of the Referee, we have rephrased parts of the abstract and the introduction to render our statements more precise.

I fully agree with the authors that in confined and/or fractal systems a linear approximation is not appropriate because some states cannot be realized and reach under such circumstance. The presented experimental data seems to be nice examples for that. Nevertheless, other numerical approaches to the low frequency density of states show that also under confinement situation becomes quadratic in the frequency dependence. See Phys. Rev. B: Condens. Matter Mater. Phys., 2010, 81, 054208. At least this paper should be cited.

The reference (Phys. Rev. B: Condens. Matter Mater. Phys., 2010, 81, 054208.) indicated by Referee 3 was already cited in the first version of our manuscript but we do agree that it deserves more elaboration. The reference mentioned, among other things, emphasizes the importance of the boundary conditions on the dynamics of the VDOS under confinement. Despite our data reported in Fig.3(a) are probably not enough to make a definitive statement nor conduct a complete analysis, the behaviour of the DOS seems compatible with the case of soft confinement reported in the reference mentioned above. The peak found in bulk ice around ~ 6 meV is gradually shifted to lower frequency and it broadens upon confinement. This seems compatible with our structure in which the GOM membrane, as a soft material, does not induce hard-type confinement.

Let us emphasize also that our setup is slightly different from the one considered in the reference above. Our work is not only focused on amorphous materials but also crystalline ones under confinement. In any case, the paper mentioned by the Referee is definitely important in relation to our analysis and it is now discussed in the revised manuscript.

In figures 2,3 and 4 at the x-axis omega is used. General in physics omega is used as a symbol for the frequency (see also equ. 2). In these figures energy values are given. This is not consistent.

We are sorry for this inconsistency. It has been fixed in the revised version of the manuscript. We now use $\hbar\omega$ (where \hbar is the Planck constant) instead of E . We thank the Referee for pointing this out.

Minor comment: In figure 7 it should be THz. Temperature differences should be given in K not in C in accordance with international regulations.

Thanks for pointing this out. We have corrected these inaccuracies in the revised manuscript.

REVIEWERS' COMMENTS

Reviewer #1 (Remarks to the Author):

Authors in the response letter have virtually addressed all my concerns raised in the initial review. The new SI materials are indeed complementing the data presented in the manuscript and the quality of the manuscript has improved significantly. I am now happy to recommend publication of the manuscript in its current form.

Reviewer #2 (Remarks to the Author):

The authors are commended for having done a very thorough job to answer to referees' questions. In particular, when addressing my question on the transition from w_2 to w_3 behaviors, the authors took on additional studies from both theory and simulation perspectives. They not only demonstrated the transition, but also found a cutoff frequency below which the scaling changes. Moreover, the theory and simulations agree with each other. I also note that the basis for the theoretical analysis has been published in PRB.

I now consider the paper on solid footing and am happy to recommend it for publication in Nature Communications.

Reviewer #3 (Remarks to the Author):

After the revision process the manuscript can be published as it is in accordance with international regulations.

Response to the Referees for manuscript NCOMMS-21-37774-A

Dear Referees,

thanks a lot for your reviews and the positive comments on our manuscript. Thanks to the suggestions of the Referees, our work has considerably improved since we first submitted it.

Referee 1

Authors in the response letter have virtually addressed all my concerns raised in the initial review. The new SI materials are indeed complementing the data presented in the manuscript and the quality of the manuscript has improved significantly. I am now happy to recommend publication of the manuscript in its current form.

We thank Referee 1 for his/her positive evaluation and recognition of our work and for recommending it for publication. The comments by Referee 1 have greatly helped us to supplement and improve SI with more simulation details, making our work more complete and convincing.

Referee 2

The authors are commended for having done a very thorough job to answer to referees' questions. In particular, when addressing my question on the transition from w_2 to w_3 behaviors, the authors took on additional studies from both theory and simulation perspectives. They not only demonstrated the transition, but also found a cutoff frequency below which the scaling changes. Moreover, the theory and simulations agree with each other. I also note that the basis for the theoretical analysis has been published in PRB. I now consider the paper on solid footing and am happy to recommend it for publication in Nature Communications.

We thank Referee 2 for the positive comments on our work and for suggesting its publication. His/her questions prompted us to improve the theoretical

analysis and to add new simulations in support of it. The integrity and persuasiveness of our work have been greatly enhanced in the process.

Referee 3

After the revision process the manuscript can be published as it is .

We thank Referee 3 for his/her positive opinion about our work and for suggesting its publication in Nature Communications. The suggestions of Referee 3 have been very helpful in improving our manuscript.